# Evodiamine Exhibits Anti-Bladder Cancer Activity by Suppression of Glutathione Peroxidase 4 and Induction of Ferroptosis

**DOI:** 10.3390/ijms24076021

**Published:** 2023-03-23

**Authors:** Che-Yuan Hu, Hung-Tsung Wu, Yan-Shen Shan, Chung-Teng Wang, Gia-Shing Shieh, Chao-Liang Wu, Horng-Yih Ou

**Affiliations:** 1Institute of Clinical Medicine, College of Medicine, National Cheng Kung University, Tainan 70101, Taiwan; greatoldhu@gmail.com (C.-Y.H.);; 2Department of Urology, National Cheng Kung University Hospital, College of Medicine, National Cheng Kung University, Tainan 70101, Taiwan; 3Department of Internal Medicine, School of Medicine, College of Medicine, National Cheng Kung University, Tainan 70101, Taiwan; 4Department of Surgery, National Cheng Kung University Hospital, College of Medicine, National Cheng Kung University, Tainan 70101, Taiwan; 5Department of Microbiology and Immunology, College of Medicine, National Cheng Kung University, Tainan 70101, Taiwan; 6Department of Urology, Tainan Hospital, Ministry of Health and Welfare, Executive Yuan, Tainan 70043, Taiwan; 7Ditmanson Medical Foundation Chia-Yi Christian Hospital, Chiayi 600566, Taiwan; 8Department of Biochemistry and Molecular Biology, College of Medicine, National Cheng Kung University, Tainan 70101, Taiwan; 9Department of Internal Medicine, National Cheng Kung University Hospital, College of Medicine, National Cheng Kung University, Tainan 70101, Taiwan

**Keywords:** evodiamine, bladder cancer, ferroptosis, glutathione peroxidase 4, lipid peroxidation

## Abstract

Evodiamine (EVO) exhibits anti-cancer activity through the inhibition of cell proliferation; however, little is known about its underlying mechanism. To determine whether ferroptosis is involved in the therapeutic effects of EVO, we investigated critical factors, such as lipid peroxidation levels and glutathione peroxidase 4 (GPX4) expression, under EVO treatment. Our results showed that EVO inhibited the cell proliferation of poorly differentiated, high-grade bladder cancer TCCSUP cells in a dose- and time-dependent manner. Lipid peroxides were detected by fluorescence microscopy after cancer cell exposure to EVO. GPX4, which catalyzes the conversion of lipid peroxides to prevent cells from undergoing ferroptosis, was decreased dose-dependently by EVO treatment. Given the features of iron dependency and lipid-peroxidation-driven death in ferroptosis, the iron chelator deferoxamine (DFO) was used to suppress EVO-induced ferroptosis. The lipid peroxide level significantly decreased when cells were treated with DFO prior to EVO treatment. DFO also attenuated EVO-induced cell death. Co-treatment with a pan-caspase inhibitor or necroptosis inhibitor with EVO did not alleviate cancer cell death. These results indicate that EVO induces ferroptosis rather than apoptosis or necroptosis. Furthermore, EVO suppressed the migratory ability, decreased the expression of mesenchymal markers, and increased epithelial marker expression, determined by a transwell migration assay and Western blotting. The TCCSUP bladder tumor xenograft tumor model confirmed the effects of EVO on the inhibition of tumor growth and EMT. In conclusion, EVO is a novel inducer for activating the ferroptosis of bladder cancer cells and may be a potential therapeutic agent for bladder cancer.

## 1. Introduction

Approximately 25% of bladder urothelial carcinomas (UCs) are muscle-invasive or metastatic [1]. Chemotherapy remains the first choice in treating metastatic bladder UCs [2]. However, almost 50% of patients with metastasis are ineligible for traditional cisplatin-based chemotherapy, and progression after first-line chemotherapy is also common [3,4]. Second-line chemotherapy, such as taxane or vinflunine, can only achieve approximately 10% of the objective response rate (ORR) [5]. On the other hand, new, subsequent therapies, such as immune checkpoint inhibitors (ICIs), provide clinical benefits in some patients with a limited response rate from 13 to 21% [6,7]. Therefore, there is an urgent need to consider patients’ comorbidities and medicine efficacy when developing an appropriate treatment strategy or combination therapy.

Some natural compounds extracted from herbs have been considered as potential adjuvants to enhance the anti-neoplastic ability of current medication [8,9]. Evodiamine (EVO), one of the quinazoline alkaloids, is an essential isolated from the seeds of Evodia rutaecarpa (Wu-Zhu-Yu) [10]. It has been demonstrated to exhibit anti-cancer activities in non-small-cell lung cancer, hepatocellular carcinoma (HCC), leukemia, and breast cancer [11,12,13,14,15]. However, the literature discussing the anti-cancer effect of EVO in bladder UC is currently limited [15,16]. It is suggested that EVO acts in synergy with a target therapy or chemotherapeutic agents when multidrug resistance occurs in tumor cells [17,18,19]. EVO causes G2/M cell cycle arrest [20] and suppresses tumor cell invasion and metastasis [21,22,23,24]. However, the precise mechanism of this multifaceted anti-cancer activity of EVO remains obscure. Previous studies have revealed that EVO exerts its cytotoxic effect by increasing the amount of reactive oxygen species (ROS) in cancer cells, and ROS can be amplified in a positive feedback loop [25,26]. Ferroptosis is an iron-dependent and lipid-peroxidation-driven form of cell death [27]. It occurs when cells encounter ambient oxidative stress, with the massive membrane lipid peroxidation compromising the selective permeability of the plasma membrane [28]. In this process, iron serves as a catalytic agent to convert peroxides into free radicals. RSL3 acts as a ferroptosis inducer by inactivating glutathione peroxidase 4 (GPX4), an anti-oxidant enzyme that protects cells against membrane lipid peroxidation [29]. GPX4 catalyzes the conversion of lipid peroxide at the expense of glutathione to prevent cells from undergoing ferroptosis.

Cancer cells are usually under constant oxidative stress, yet ferroptosis seldom occurs due to the delicate equilibrium between catalytic irons and thiols [30]. Over the last decade, the value of inducing ferroptosis, as well as its linkage with cancer therapy, has been noticed. Some studies investigated the pivotal role of ferroptosis in causing cancer cell death. Others explored the combination of inducing ferroptosis with traditional chemotherapy to synergize antitumor effects [31]. Furthermore, due to its attribute of programmed necrosis, ferroptosis is predicted to be more immunogenic than apoptosis. This makes a ferroptosis inducer a possible immuno-enhancer in this era of immunotherapy [32]. In the current study, we demonstrate that EVO decreases GPX4 expression to facilitate lipid peroxidation in UC cells. For the first time, our results reveal that EVO suppresses UC cell proliferation by inducing cell death through ferroptosis. It also disrupts cell epithelial–mesenchymal transition (EMT) to attenuate migratory ability.

## 2. Results

### 2.1. EVO Induces Cytotoxicity and Promotes Cell Cycle Arrest at the G2/M Phase in TCCSUP Cells

First, we evaluated the effects of EVO on the cell viability of human bladder cancer cells with the tetrazolium salt MTS and the electron coupling reagent phenazine ethosulfate (PES). The MTS formazan product is soluble in tissue culture medium, eliminating the solubilization steps normally required for the MTT assay in which insoluble formazan crystals are formed [33]. The colorimetric MTS assay shows that EVO induced a cytotoxic effect in a dose- and time-dependent manner in TCCSUP cells (Figure 1A,B). While control cells became near confluent after 24 h in the culture, EVO dose-dependently induced cell death in TCCSUP cells as observed using phase-contrast microscopy (Figure 1C). As cell cycle arrest may play an essential role in the inhibition of cell proliferation, we assessed the effects of EVO on cell cycle progression. The DNA content of the TCCSUP cells was measured by flow cytometric analysis after their exposure to EVO for 24 h. DNA histograms demonstrate that treatment with EVO increased the cell population at the G2/M phase, while it decreased the percentage of G1 cells (Figure 1D). Compared with the control cells, EVO treatment resulted in a dose-dependent accumulation of cells in the G2/M phase, accompanied by a decrease in the G1 phase in TCCSUP cells (Figure 1E). Collectively, these results indicate that treatment with EVO induces cytotoxicity and G2/M cell cycle arrest in TCCSUP cells.

### 2.2. EVO Suppresses GPX4 Expression to Facilitate Lipid Peroxidation in TCCSUP Cells

Next, we further explored whether EVO-induced cell death was attributed to ferroptosis. Given that ferroptosis is characterized by the accumulation of reactive oxygen species (ROS) from iron metabolism [34], we detected ROS production by fluorescence microscopy with 2′,7′-dichlorodihydrofluorescein diacetate (DCFDA) staining. DCFDA is a cell-permeable, redox-sensitive fluorescent probe that is oxidized by ROS to yield green fluorescence. As ferroptosis is an iron-dependent form of oxidative cell death, it can be inhibited by the iron chelator deferoxamine (DFO), a medication used for treating iron toxicity. Figure 2A shows that ROS production was detected in the cells after exposure to EVO. However, the ROS level significantly decreased when cells were pretreated with DFO before EVO exposure. In contrast, cells treated with DFO alone had no impact on ROS production. As ferroptosis is driven by the lethal accumulation of lipid peroxides in the cell membrane [34], determination of the lipid peroxide amount in the cell membrane with the lipid-peroxidation-sensitive dye C11-BODIPY and flow cytometry can be used to detect ferroptosis. In TCCSUP cells, EVO induced significant lipid ROS accumulation, as indicated by a dramatic increase in the amount of oxidized C11-BODIPY (green fluorescence) (Figure 2B,C). Furthermore, EVO-induced lipid peroxide accumulation could be alleviated by prior treatment with DFO (Figure 2B,C). Since the antioxidant enzyme GPX4, which serves as the master regulator of ferroptosis, catalyzes lipid peroxides to protect cells from ferroptosis, we then examined whether EVO treatment affected GPX4 expression in TCCSUP cells. Treatment with EVO significantly reduced the expression of GPX4 (Figure 2D,E), suggesting that GPX4 downregulation may play a pivotal role in EVO-induced ferroptosis in bladder cancer cells. Taken together, these results demonstrate that EVO induces ferroptosis in bladder cancer cells, as evidenced by increases in ROS production and lipid peroxidation and the suppression of GPX4.

### 2.3. EVO Induces G2/M Cell Cycle Arrest and Ferroptosis in TCCSUP Cells

Having shown that the level of cellular lipid peroxides was reduced in TCCSUP cells after DFO treatment and prior to EVO exposure, we further used inhibitors of ferroptosis, apoptosis, and necroptosis to elucidate the involvement of ferroptosis in EVO-induced cell death in TCCSUP cells. Treatment with DFO reversed EVO-induced cell death (Figure 3A). Notably, combination treatment with DFO and EVO significantly improved cell survival compared with single treatment with EVO (Figure 3B). The cell viability did not differ between cells treated with DFO alone and untreated control cells, indicating that DFO is non-cytotoxic to TCCSUP cells (Figure 3B). In addition, co-treatment with the pan-caspase inhibitor Z-VZD-FMK and EVO had no impact on EVO-induced cell death (Figure 3B), suggesting that apoptosis is not primarily involved in EVO-induced cell death. The pro-necrotic kinases in the receptor-interacting protein (RIP) family are critical modulators of programmed necrosis, such as necroptosis [35]. Necrostatin-1 targets RIP1/receptor-interacting protein kinase 1 (RIPK1) and acts as a specific inhibitor of necroptosis. We therefore co-treated cells with necrostatin-1 and EVO to examine whether necroptosis participated in EVO-induced cell death. No significant difference in cell viability was observed between the EVO and EVO plus necrostatin-1 groups, suggesting that necroptosis is not involved in EVO-induced cell death (Figure 3B). An analysis of the cell cycle using propidium iodide (PI) staining and flow cytometry revealed that EVO induced cell cycle arrest in the G2/M phase, which could be alleviated by DFO treatment (Figure 3C,D). These results collectively indicate that EVO induces G2/M cell cycle arrest and promotes lipid-oxidation-dependent ferroptosis in TCCSUP cells. Furthermore, such effects can be reversed by the ferroptosis inhibitor DFO but not by the pan-caspase inhibitor Z-VZD-FMK or the necroptosis inhibitor necrostatin-1.

### 2.4. EVO Inhibits Cell Migration and EMT in TCCSUP Cells

In addition to ferroptosis, we also evaluated the effects of EVO on cell migration and EMT in TCCSUP cells. To exclude the possibility that the effect of EVO on cell migration may be confounded by its cytotoxic effect, particularly at high doses, we used low concentrations of EVO (5–15 µM) to avoid evident cytotoxicity caused by EVO in the cell migration assay. TCCSUP cells cultured in 10 cm culture dishes were treated with or without EVO for 24 h. The treated cells were then trypsinized, resuspended in the culture medium in the absence of EVO, reseeded at 1 × 10^4^/well in the upper chamber of the transwell in 24 well plates and 96 well plates, and cultured for an additional 24 h for assessing cell migration and cell viability, respectively. Figure 4A shows representative microscopic images of the migratory cells with or without EVO pretreatment. The number of migratory cells decreased in EVO-treated cells compared with that of the control cells. Furthermore, the quantitative analysis of migratory cells revealed that EVO inhibited cell migration in a dose-dependent manner (Figure 4B). The cell viability of EVO-pretreated cells was concurrently assessed by the MTS assay. Appendix A shows that at concentrations of 10 and 15 µM, EVO only decreased cell viability by 10% and 15%, respectively, when compared to the control cells. In addition, to exclude the minor effect of EVO-induced cytotoxicity on cell migration, we normalized the number of migratory cells (Figure 4B) with the percentage of cell viability (Appendix A) by dividing the migratory cell number with the cell viability percentage. The degree of EVO-induced inhibition of cell migration remained similar after the normalization of cell numbers (Appendix A). These results clearly indicate that EVO inhibits cell migration independent of its cytotoxic effect. Since the EMT plays an important role in cancer progression and metastasis, we next determined whether EVO impacted the expression of EMT-related proteins (E-cadherin, N-cadherin, vimentin, and Snail) in TCCSUP cells. Results from Western blotting (Figure 4C) and the quantitative analysis (Figure 4D–G) showed that expression of the mesenchymal markers N-cadherin, vimentin, and Snail decreased, whereas the epithelial marker E-cadherin increased in cells treated with EVO. Taken together, these results demonstrate that EVO inhibits bladder cancer cell migration by suppressing EMT.

### 2.5. EVO Inhibits Tumor Growth and Induces Mesenchymal-Epithelial Transition (MET) in NOD-SCID Mice Bearing TCCSUP Tumor Xenografts

Given that EVO induced cell death in TCCSUP cells by promoting cell cycle arrest, inhibiting cell migration, and suppressing EMT in vitro, we next used the TCCSUP tumor xenograft model in immunodeficient mice to investigate whether EVO exerted antitumor effects in vivo. TCCSUP cells were subcutaneously implanted in the right flank of NOD-SCID mice, followed by an intraperitoneal injection of EVO (20 mg/kg) or PBS for 35 consecutive days. Tumor growth was monitored weekly in tumor-bearing mice after treatment with EVO or PBS, and the mice were sacrificed at day 35. Gross appearances of day 35 tumors are shown in Figure 5A. Tumor sizes were remarkably reduced in the mice treated with EVO compared with those treated with PBS. Figure 5B shows that treatment with EVO significantly decreased the tumor size compared with PBS treatment in tumor-bearing mice. Furthermore, EVO treatment increased the expression of E-cadherin (Figure 5D), but decreased the expression of N-cadherin (Figure 5E), vimentin (Figure 5F), and Snail (Figure 5G) in the tumor tissues of mice receiving EVO treatment. Taken together, these results indicate that EVO inhibits tumor growth and induces MET in vivo, suggesting the therapeutic potential of EVO for the treatment of bladder UC.

## 3. Discussion

In the present study, we demonstrate that EVO induces G2/M cell cycle arrest in human bladder cancer cells. Furthermore, EVO exerts cytotoxic effects and inhibits EMT in vitro and in vivo. These activities are associated with ferroptosis and the downregulation of GPX4. GPX4 is a selenocysteine-containing enzyme that prevents cells from undergoing ferroptosis by reducing lipid hydroperoxides to their corresponding lipid alcohols [36]. We present a schematic illustration showing EVO-induced ferroptosis through the suppression of GPX4 in human bladder cancer cells (Figure 6). A previous study showed that RSL3, a specific ferroptosis inducer [28], bound to and inactivated GPX4 [36]. COH-BR1 breast cancer cells co-cultured with 7α-cholesterol hydroperoxide showed that GPX4 reduced lipid peroxide in a glutathione-dependent manner, and treatment with RSL3 inactivated GPX4 and suppressed its peroxide reduction [36]. However, treatment with RSL3 in mice that had been inoculated with engineered tumorigenic BJeLR cells suppressed tumor growth. Moreover, the tumor features resembled GPX4-dysregulated ferroptosis, with a representative marker of ferroptosis [36]. Recent reports have demonstrated that in a high-mesenchymal cell state, cancer cells might be more dependent on GPX4 [37]. Zinc finger E-box-binding homeobox 1 (ZEB-1) is one of the EMT regulators. Among 610 cancer cell lines, those with high ZEB-1 transcription levels were remarkably more sensitive to a GPX4 inhibitor [37]. In KP4 high-mesenchymal state pancreatic cancer cells, ZEB-1 knockdown prevented the cells from GPX4-inhibitor-induced cell death [37]. Furthermore, cell lines in a mesenchymal state were inclined towards ferroptosis if GPX4 was knocked out, whereas epithelial-state GPX4-knockout cells stayed unaffected [37]. This may explain why EVO can suppress GPX4 expression in TCCSUP cells and subsequently inhibit their mesenchymal transition and migratory ability.

High-grade or drug-resistant cancer cells are also selectively sensitive to GPX4 inhibitors [38]. Various drug-tolerant “persister” cells in residual melanoma, breast, lung, or ovarian cancers share a common feature of vulnerability to ferroptosis compared with their parental cells [38]. In the A375 melanoma xenograft model, the residual tumor from GPX4 wild-type tumors relapsed after treatment, while the GPX4-deficient tumor did not. These results suggest that GPX4 is essential for maintaining cell growth in residual tumors in vivo [38]. Therefore, co-treatment with a GPX4 inhibitor and current standard medications might provide patient benefits by decreasing the number of persistent cancer cells. TCCSUP cells have been demonstrated to be a poorly differentiated, high-grade bladder cancer cell line. The gemcitabine-resistant TCCSUP subline showed resistance to multiple chemotherapeutic agents, such as cisplatin, paclitaxel, vinflunine, methotrexate, oxaliplatin, 5-fluorouracil, and doxorubicin [39]. To the best of our knowledge, our study is the first report to demonstrate that EVO can induce cell death through ferroptosis accompanied by suppression of GPX4.

RSL3 serves as a GPX4 inhibitor and relies on its activated alkyl chloride to bind the selenocysteine residue of GPX4. However, such GPX4 inhibitors, which are dependent on their chloroacetamide group, are not satisfied by their promiscuity and instability, resulting in low bioavailability [40]. Using masked precursors that undergo chemical transformations intracellularly is a better strategy for targeting GPX4 [40]. EVO is susceptible to P450-enzyme-regulated dehydrogenation and then generates electrophilic intermediates [41]. The non-covalently-bonded EVO–phospholipid complex has also been designed to enhance its biological efficacy [42]. The relative bioavailability of the complex significantly increased to over 200% of free EVO [42]. In addition, compared with pure EVO, the Wu-Zhu-Yu extract had better EVO bioavailability, indicating that some components in Wu-Zhu-Yu may change the pharmacokinetics of the drug [43]. Overall, Wu-Zhu-Yu or its main bioactive ingredient, EVO, acts as a potential adjuvant medication to enhance the efficacy of target therapy or chemotherapy, thereby preventing high-grade cancer cell proliferation or transitions to a mesenchymal state.

Non-coding RNAs, such as microRNA (miRNA) and long non-coding RNA (lncRNA), play pivotal roles in gene regulation and biological processes in both health and disease, including cancer initiation and progression. Therefore, they are recognized as potential diagnostic and prognostic markers as well as therapeutic targets for cancers, including bladder UC [44]. Increasing evidence has indicated that ferroptosis is associated with cancer initiation, progression, and suppression [45]. Furthermore, miRNAs can either promote or suppress tumors through regulating ferroptosis by interfering with iron metabolism, lipid peroxidation, and the system Xc^-^/GPX4 axis [46,47]. EVO can affect the expression of non-coding RNAs and their target genes in cancer cells, suggesting that EVO can mediate gene expression at the post-transcriptional level. Accumulating evidence has revealed that miRNAs are involved in the control of DNA methylation machinery [48,49,50]. Huang et al. reported that EVO increases the expression of miR-152, miR-429, and miR-29a, which in turn downregulates DNA methyltransferase 1 (DNMT1), DNMT3A, and DNMT3B in colorectal cancer cells, resulting in reversing the epigenetic silencing of tumor suppressor genes and inhibiting cancer cell growth [51]. Apart from DNA methylation, the three miRNAs possess functions as tumor suppressors. In human hepatocellular carcinoma tissues, miRNA-152 levels are reduced, and transferrin receptor 1 (TFR1) is overexpressed. Furthermore, miR-152 regulates iron homeostasis by downregulating TFR1, thereby participating in ferroptosis in liver cancer cells [52]. The miR-200 family is highly expressed in epithelial cells and plays a crucial role in maintaining the epithelial phenotype, which is achieved by inhibiting the expression of factors that promote EMT [53]. MiR-429 belongs to the miR-200 family, suggesting that it may have roles in EMT suppression [53,54]. Moreover, miR-29a inhibits EMT in lacrimal gland adenoid cystic carcinoma and endometrial cancer [55,56]. LncRNAs have been regarded as some of the key regulators in cancer progression and drug resistance by regulating the expression of downstream genes and various biological processes. The silencing of lncRNA NEAT1 can enhance erastin-mediated ferroptosis in non-small-cell lung cancer cells by increasing intracellular lipid peroxidation, indicating the involvement of NEAT1 in the regulation of ferroptosis sensitivity [57]. NEAT1 plays a tumor-promoting role in ovarian cancer. It was recently shown that EVO decreases the expression of NEAT1 and CDK19, but increases miR-152 expression in ovarian cancer cells [58]. Furthermore, EVO attenuates ovarian cancer cell progression via the NAET1/miR-152-3p/CDK19 axis [58]. In conclusion, EVO can upregulate tumor-suppressor miRNAs and downregulate oncogenic lncRNA NEAT1, providing molecular mechanisms linking EVO to non-coding RNAs that can contribute, in part, to its anti-cancer activities.

A limitation of our study is the lack of a detailed molecular mechanism of the way in which EVO suppresses GPX4 expression. Additional research is needed to determine whether the mechanism by which EVO-mediated GPX4 suppression involves non-coding RNAs and to clarify whether the GPX4 suppression and/or modulation of non-coding RNAs is responsible for the observed inhibition of EMT in bladder UC. We have previously reported that EVO activates WW-domain-containing oxidoreductase (WWOX) to exert its anti-cancer efficacy [12]. The association between WWOX upregulation and GPX4 downregulation under EVO exposure is currently under investigation.

## 4. Materials and Methods

### 4.1. Cell Culture

The human bladder cancer TCCSUP cell line was purchased from the Bioresource Collection and Research Center (Food Industry Research and Development Institute, Hsinchu, Taiwan). Cells were cultured in high-glucose Dulbecco’s modified Eagle’s medium (DMEM; Hyclone, South Logan, UT, USA) supplemented with 10% fetal bovine serum (Hyclone, Logan, UT, USA) and 1% penicillin and streptomycin at 37 °C in an atmosphere of 5% CO_2_.

### 4.2. Cell Proliferation Assay

Cell proliferation was determined to evaluate the EVO-induced cytotoxic effect on TCCSUP cells with an MTS assay, using the CellTiter 96 AQueous One Solution cell proliferation assay kit (Promega, Madison, WI, USA) according to manufacturer’s instructions. TCCSUP cells were seeded at 5 × 10^3^ cells/well in 96 well plates and cultured for 24 h. Cells were then treated with various concentrations of EVO (Sigma-Aldrich, St. Louis, MO, USA) for 24 h or with 15 μM EVO for different intervals of time. Subsequently, the MTS/PES solution (20 μL) provided by the kit was added to each well, and the cells were incubated at 37 °C for 2 h. The optical absorbance at 490 nm was measured with a Multiskan GO microplate spectrophotometer (Thermo Scientific, Waltham, MA, USA).

### 4.3. Transwell Migration Assay

Cell migration was measured using 24 well transwells with inserts containing 8 μm membrane filters (Merck Millipore, Burlington, MA, USA). TCCSUP cells were seeded at 5 × 10^5^ cells/dish in 10 cm dishes and cultured for 24 h. Cells were then treated with EVO or the vehicle DMSO for an additional 24 h. The treated cells were then trypsinized and resuspended in the culture medium in the absence of EVO. Subsequently, the cell suspension (200 μL) containing 1 × 10^4^ cells was added into the upper chamber of the transwells, which were then placed in 24 well plates to which the culture medium (600 μL) was added. After being cultured at 37 °C for 24 h, cells that migrated through the membrane to the lower surface were fixed, stained with Giemsa’s azure eosin methylene blue solution (Merck Millipore), and quantified.

### 4.4. Detection of Cellular ROS Generation and Lipid Peroxidation

Cellular ROS generation was measured using the ROS-sensitive fluorometric probe DCFDA (Invitrogen, Carlsbad, CA, USA). TCCSUP cells were seeded at 5 × 10^5^ cells/dish in 10 cm dishes and cultured for 24 h. Subsequently, cells were pretreated with the iron chelator DFO (Cayman Chemical Company, Ann Arbor, MI, USA) or the vehicle DMSO for 1 h, followed by treatment with EVO or the vehicle DMSO for 24 h. The treated cells were stained with 10 μM DCFDA for 30 min at 37 °C in the dark. Finally, ROS-producing cells were detected under fluorescence microscopy (Olympus, Tokyo, Japan) for cells with green fluorescence. Nuclei were counterstained with DAPI (blue). To detect membrane lipid peroxidation, TCCSUP cells that had been treated with DFO for 1 h and EVO for 24 h were incubated with 1 μM C11-BODIPY (Cayman Chemical) at 37 °C for 30 min in the dark. The amount of lipid peroxides (green fluorescence) was measured by flow cytometry using CytoFLEX (Beckman Coulter, Brea, CA, USA).

### 4.5. Cell Cycle Analysis

TCCSUP cells cultured in 10 cm dishes were treated with various concentrations of EVO for 24 h with or without prior DFO (100 µM) treatment for 1 h. Subsequently, the treated cells were harvested, washed with PBS, and fixed in cold 70% ethanol for 30 min at 4 °C. The fixed cells were washed with PBS and then treated with ribonuclease A (Cayman Chemical) and stained with PI (Cayman Chemical) for cell cycle analysis. The DNA content of the cells at different cell cycle phases was measured by flow cytometry (CytoFLEX).

### 4.6. Western Blot Analysis

Proteins were extracted by a radioimmunoprecipitation lysis buffer (VWR Chemicals, Radnor, PA, USA) with protease and phosphatase inhibitors (MedChem Express, Monmouth Junction, NJ, USA). After centrifugation at 13,000 rpm for 10 min at 4 °C, the supernatant was collected to determine protein concentrations using the bicinchoninic acid (BCA) assay (Visual Protein, Taipei, Taiwan). Samples containing 30 μg protein were separated by sodium dodecyl sulfate–polyacrylamide gel electrophoresis and then transferred to polyvinylidene difluoride membranes (Biomate, Taipei, Taiwan). The membranes were blocked with 10% skim milk in TBS-T (20 mM Tris (pH 7.4), 150 mM NaCl, and 0.1% Tween 20) for 1 h at room temperature and then incubated with primary antibodies, including antibodies against E-cadherin (IReal Biotechnology, Hsinchu, Taiwan), N-cadherin (IReal Biotechnology), vimentin (IReal Biotechnology), Snail (IReal Biotechnology), GPX4 (Proteintech, Chicago, IL, USA), and β-actin (Merck Millipore) at 4 °C overnight. The blots were then washed with TBS-T and incubated with horseradish peroxidase-conjugated secondary antibodies at room temperature for 1 h. The blots were detected by enhanced chemiluminescence (Millipore), and the relative signal intensity was quantified using the ImageJ software (U.S. National Institute of Health, Bethesda, MD, USA).

### 4.7. Animal Experiments

The experimental protocol adhered to the rules of the Animal Protection Act of Taiwan and was approved by the Institutional Animal Care and Use Committee (IACUC) of NCKU (IACUC approval numbers: 110162). Groups of 14 six-week-old male NOD/SCID mice were inoculated subcutaneously with TCCSUP cells (1 × 10^6^) at day 0 over the right flank, followed by the intraperitoneal injection of EVO (20 mg/kg) or the vehicle PBS for 35 consecutive days. Tumor volumes were measured weekly and calculated as: length × width^2^ × 0.45. Tumor size was expressed as a percentage of the tumor volume at week 1.

### 4.8. Statistics

Statistical tests were performed using GraphPad Prism (version 9.0, GraphPad software, San Diego, CA, USA). Data are expressed as the mean ± standard error of the mean (SEM). Statistical analysis was conducted using Student’s *t* test or a one-way ANOVA with Tukey’s post hoc test. Any *p* value of <0.05 was considered statistically significant.

## 5. Conclusions

In the present study, we show for the first time that EVO is a novel inhibitor capable of suppressing GPX4 expression and inducing ferroptosis in bladder UC cells. Our in vitro and in vivo studies suggest that EVO may be a potential therapeutic agent for bladder cancer.

## Figures and Tables

**Figure 1 ijms-24-06021-f001:**
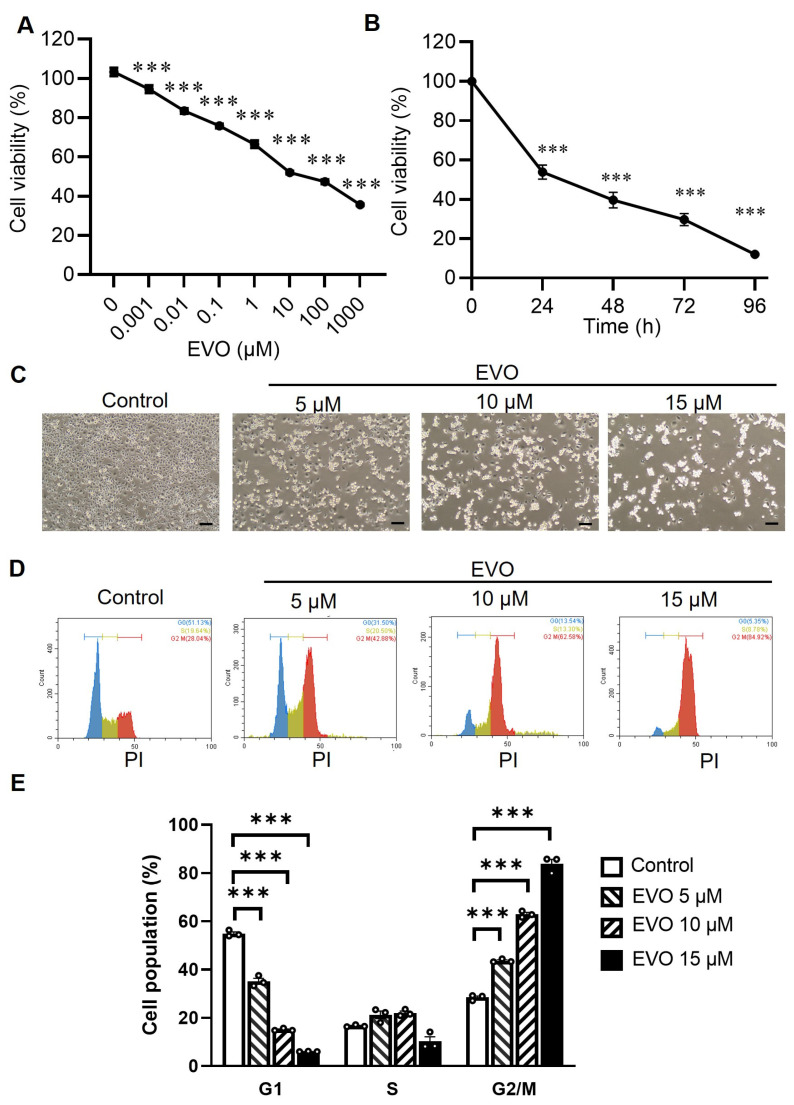
EVO induces cytotoxicity and G2/M cell cycle arrest in TCCSUP cells. (**A**) TCCSUP cells were treated with different concentrations of EVO for 24 h, and cell viability was detected by the MTS assay (n = 8). (**B**) Time course of cell viability in response to 15 μM EVO, analyzed by the MTS assay (n = 8). (**C**) Representative phase-contrast microscopic images of EVO-induced cell death are shown (Scale bar = 100 μm). (**D**) Cell cycle analysis through PI staining, followed by flow cytometric analysis for the cells after EVO treatment for 24 h. DNA histograms of cells exposed to various concentrations of EVO are shown. (**E**) Quantitative measurement of the cell population at different phases of the cell cycle (n = 3 for each group). Percentages of cell population at G1, S, and G2/M phases are shown. Data are shown as the mean ± SEM. *** *p* < 0.001 as compared with the control group.

**Figure 2 ijms-24-06021-f002:**
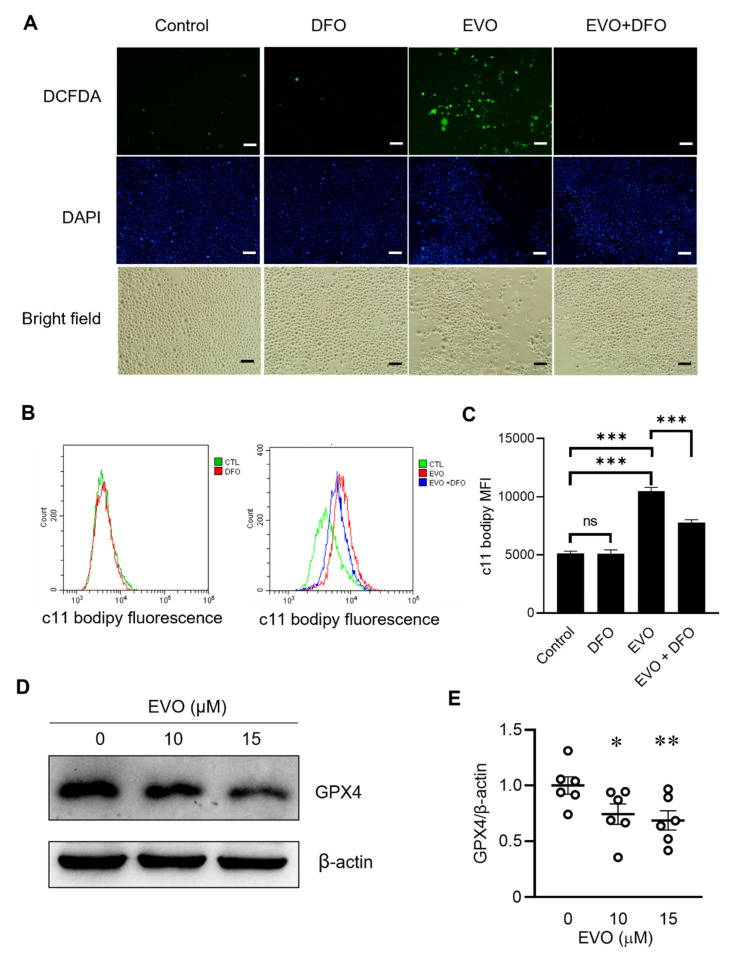
EVO decreases GPX4 expression to facilitate lipid peroxidation in TCCSUP cells. (**A**) TCCSUP cells were treated with 100 µM of the iron chelator DFO or the vehicle (DMSO) for 1 h, followed by treatment with 15 µM EVO or the vehicle (DMSO) for 24 h. The cells were incubated with the ROS-sensitive fluorometric probe DCFDA (10 μM) for 30 min in the dark, and the content of ROS was detected with fluorescence microscopy. Nuclei were stained with DAPI (blue). Representative fluorescent and bright-field microscopic images are shown (scale bar = 20 μm). (**B**,**C**) TCCSUP cells were treated with DFO or DMSO for 1 h, followed by treatment with EVO or DMSO for 24 h as described in A. The cells were then stained with the lipid peroxidation reporter probe C11-BODIPY (1 μM) at 37 °C for 30 min in the dark. The amount of lipid peroxides (green fluorescence) was measured by flow cytometry (n = 4 for each group). Fluorescence histograms (**B**) and mean fluorescence intensity (MFI) values (**C**) are shown. Data are mean ± SEM. CTL—control; ns—non-significant. (**D**,**E**) Detection (**D**) and quantitation (**E**) of the anti-oxidant GPX4 protein in TCCSUP cells after treatment with EVO for 24 h by Western blot analysis (**D**) and densitometric analysis using the ImageJ software (n = 6 for each group) (**E**). Expression of β-actin served as the loading control. The horizontal bars shown in F denote the mean value of each group. * *p* < 0.05, ** *p* < 0.01, *** *p* < 0.001 as compared with the control group.

**Figure 3 ijms-24-06021-f003:**
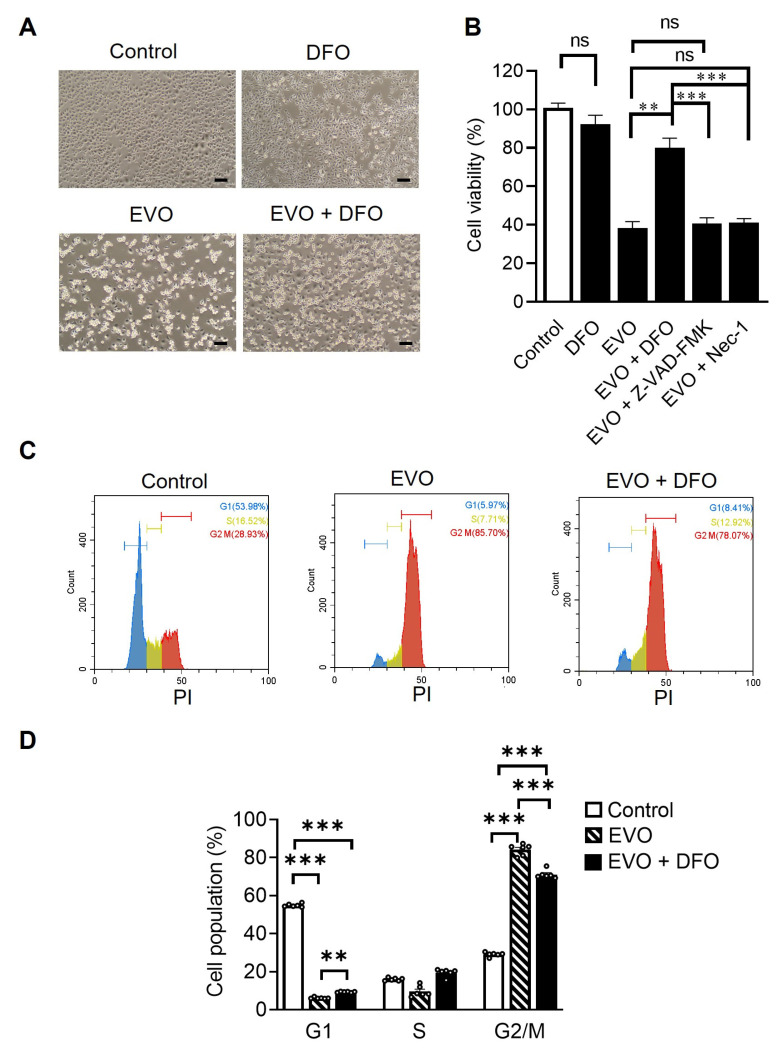
EVO induces cell death and cell cycle arrest through ferroptosis in TCCSUP cells. (**A**) TCCSUP cells were treated with 100 µM DFO or the vehicle (DMSO) for 1 h, followed by treatment with 15 µM EVO or the vehicle (DMSO) for 24 h. Representative phase-contrast microscopic images of EVO-induced cell death are shown (Scale bar = 100 μm). (**B**) TCCSUP cells were treated with DFO (100 µM), Z-VAD-FMK (20 µM), necrostatin-1 (Nec-1, 50 µM), or the vehicle (DMSO) for 1 h, followed by treatment with 15 µM EVO or the vehicle (DMSO) for 24 h. Cell viability was analyzed by the MTS assay (n = 8 for each group). (**C**,**D**) Cell cycle analysis with PI staining and flow cytometry after EVO treatment for 24 h with or without prior DFO treatment for 1 h. DNA content histograms (**C**) and quantitation (**D)** of different cell cycle phases (n = 6 for each group) are shown. Data are mean ± SEM. ** *p* < 0.01, *** *p* < 0.001 as compared with the control group.

**Figure 4 ijms-24-06021-f004:**
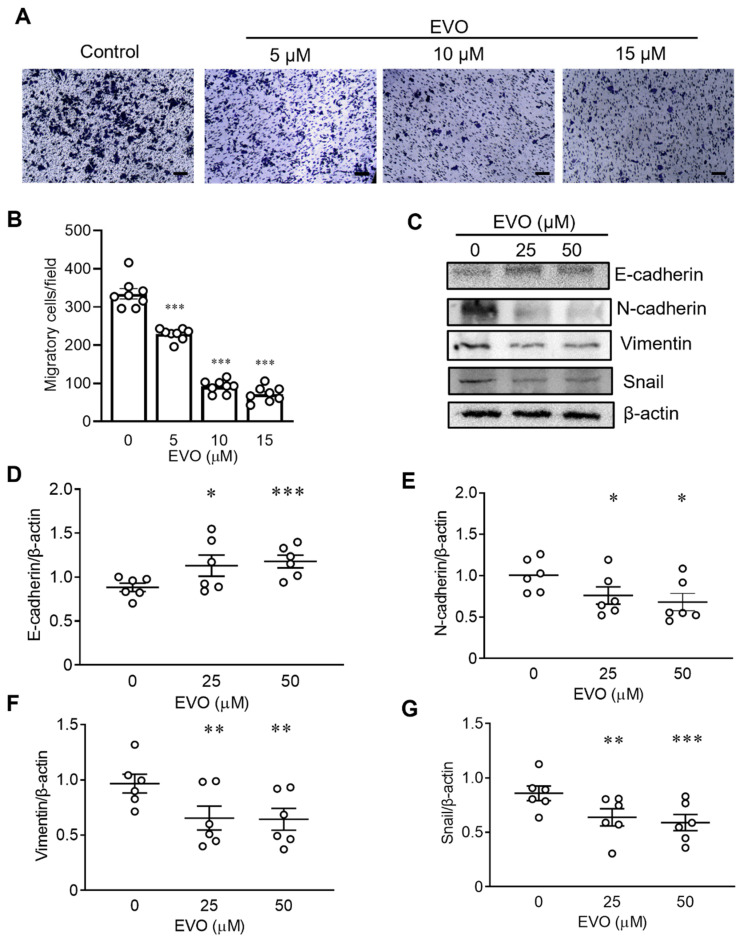
EVO inhibits cell migration and EMT in TCCSUP cells. (**A**,**B**) TCCSUP cells that were treated with indicated concentrations of EVO for 24 h were seeded (1 × 10^4^ cells) in the upper chamber of the transwell and cultured for 24 h. Cells that migrated through the membrane to the lower surface were stained with Giemsa and quantified. Representative microscopic images (magnification × 100; scale bar = 100 μm) (**A**) and quantitation (n = 8 for each group) (**B**) of the migratory cells are shown. (**C**–**G**) Detection (**C**) and quantitation of E-cadherin (**D**), N-cadherin (**E**), vimentin (**F**), and Snail (**G**) proteins in TCCSUP cells after treatment with indicated concentrations of EVO for 24 h by Western blot analysis (**C**) and densitometric analysis using the ImageJ software (n = 6 for each group) (**D**–**G**). Expression of β-actin served as the loading control. Data are mean ± SEM. The horizontal bars shown in (**D**–**G**) denote the mean value of each group. * *p* < 0.05, ** *p* < 0.01, *** *p* < 0.001 as compared with the control group.

**Figure 5 ijms-24-06021-f005:**
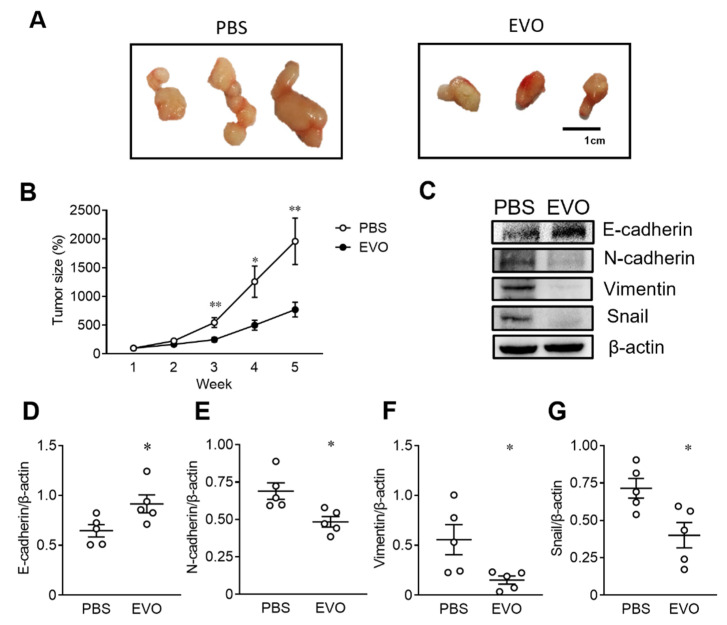
EVO inhibits tumor growth and induces MET in NOD-SCID mice bearing TCCSUP tumor xenografts. (**A**–**G**) Groups of 14 NOD/SCID mice were inoculated subcutaneously with TCCSUP cells (1 × 10^6^) at day 0, followed by intraperitoneal injection of EVO (20 mg/kg) or PBS for 35 consecutive days. Representative images of gross appearance of tumors excised at day 35 (**A**). Tumor volumes were measured weekly for 5 weeks and expressed as a percentage of the tumor volume at day 7 in each group (**B**). Detection (**C**) and quantitation of E-cadherin (**D**), N-cadherin (**E**), vimentin (**F**), and Snail (**G**) proteins in day 35 tumor tissues by Western blot analysis (**C**) and densitometric analysis using the ImageJ software (n = 5 for each group) (**D**–**G**). Expression of β-actin served as the loading control. Data are mean ± SEM. * *p* < 0.05, ** *p* < 0.01 as compared with the control group.

**Figure 6 ijms-24-06021-f006:**
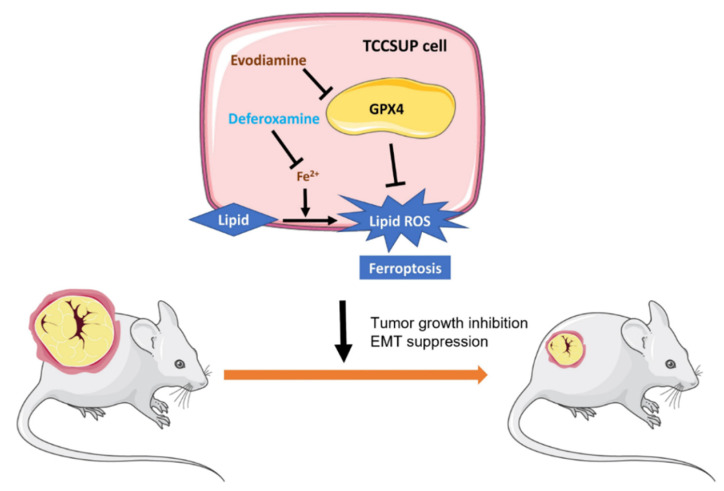
A schematic illustration of evodiamine (EVO)-induced ferroptosis through suppression of GPX4 in human bladder cancer cells. In TCCSUP human bladder cancer cells, EVO induces cell death and G2/M cell cycle arrest and inhibits cell migration and EMT. Furthermore, EVO suppresses the expression of GPX4, an antioxidant enzyme serving as a crucial negative regulator of ferroptosis, leading to the induction of ferroptosis through accumulation of lipid ROS. The iron chelator deferoxamine (DFO) alleviates EVO-induced cell death, G2/M cell cycle arrest, and lipid peroxidation via chelation of intracellular iron, which is essential for the execution of ferroptosis. In the TCCSUP tumor xenograft model, EVO inhibits tumor growth and suppresses EMT.

## Data Availability

The data presented in this study are available upon request from the corresponding author.

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
