# Peer review of "Evodiamine Exhibits Anti-Bladder Cancer Activity by Suppression of Glutathione Peroxidase 4 and Induction of Ferroptosis"

_ijms, 2023, doi:10.3390/ijms24076021_

Round 1

Reviewer 1 Report

Summary:

The authors aim to understand the mechanisms of cytotoxicity involved in bladder cancer through the treatment of Evodiamine (EVO). The authors determine that ferroptosis is the mechanism of cytotoxicity when TCCSUP cells (high grade bladder cancer) are treated with EVO. The authors properly designed the experiments to assess reactive oxygen species production, lipid peroxidation, and through treatment of an iron chelator and other controls, linked ferroptosis as the mechanism of cytotoxicity. Other experiments included GPX4 expression, which when decreased sensitize cells to ferroptosis and the increase of epithelial markers after treatment with EVO, suggesting mesenchymal-epithelial transition (MET). The authors also sought to assess the invasive potential in a transwell assay after treatment with EVO. Finally, the authors performed in vivo TCCSUP xenografts in NOD-SCID mice in which treatment with EVO significantly reduced tumor size and induced MET.

General Comments:

1.       Figure 1C – Authors should define the scale of the bar, either in the figure panel or in the figure legend

2.       Lines 112-117 – Authors need to clarify the correctly used reagent for ROS detection, which is DCFDA and was not used for lipid peroxides. They had stated this in the figure legend and materials and methods.

3.       Lines 117-119 – Define the use of C11-BODIPY as a lipid peroxidation stain to increase clarity

4.       Figure 2A – Define scale bar and increase size of it in the figure panel.

5.       Figure 2B and 2C – The authors need to confirm that there is no mistake in the selection of the data for these panels. They show close to the same results, it is possible the authors meant to show CTL and DFO alone in Figure 2B. In Panel C, the Y-axis title is visibly cropped.

6.       Line 127 – Authors should specify what are the indicated timepoints.

7.       Figure 3

a.       Define scale bar in panel A

b.      The authors should include viability data for DFO alone treatment.

8.       Figure 4

a.       Panel A and B: The reduction in cell invasion may be confounded by cell death. The authors data in Figure 1B suggests a near 50% loss in viability of TCCSUP cells when treated with 30 µM EVO after 24 h, which is the same timepoint used for the migration assay. In addition, they only indicate a significant reduction in invasive capacity at 60 µM

                                                               i.      My suggestion to the authors is to optimize the number of cells seeded for the transwell assay to allow for a longer period of measurement (48 and/or 72 h). This may allow for functional observations at lower EVO doses. However, viability assays need to be performed first in which, for example, 15 µM is shown to not significantly affect viability after 48 or 72 h.

9.       Materials and Methods

a.       MTT Assay, lines 282 and 283. What do the authors mean by removing MTT reagent?

                                                               i.      The MTT assay is based on the production of insoluble formazan crystals by live cells, if these are disturbed or removed, it may confound the assay. Typically, a solvent is added directly to the wells after the cells are incubated in MTT reagent.

                                                             ii.      I recommend the authors to confirm their viability results with other viability assays to increase the robustness of their results.

Author Response

Summary:

The authors aim to understand the mechanisms of cytotoxicity involved in bladder cancer through the treatment of Evodiamine (EVO). The authors determine that ferroptosis is the mechanism of cytotoxicity when TCCSUP cells (high grade bladder cancer) are treated with EVO. The authors properly designed the experiments to assess reactive oxygen species production, lipid peroxidation, and through treatment of an iron chelator and other controls, linked ferroptosis as the mechanism of cytotoxicity. Other experiments included GPX4 expression, which when decreased sensitize cells to ferroptosis and the increase of epithelial markers after treatment with EVO, suggesting mesenchymal-epithelial transition (MET). The authors also sought to assess the invasive potential in a transwell assay after treatment with EVO. Finally, the authors performed in vivo TCCSUP xenografts in NOD-SCID mice in which treatment with EVO significantly reduced tumor size and induced MET.

Response: We thank the reviewer’s comments and revised our manuscript accordingly.

General Comments:

  1. Figure 1C – Authors should define the scale of the bar, either in the figure panel or in the figure legend

Response: Thank you for your kind reminder. We have added the scale bar in the image and defined the scale of the bar in the legend of Figure 1 (line 116).

  1. Lines 112-117 – Authors need to clarify the correctly used reagent for ROS detection, which is DCFDA and was not used for lipid peroxides. They had stated this in the figure legend and materials and methods.

Response: Thank you for your helpful comment. We have corrected our mistake and revised the text in the results section as follows (lines 124-131): “To further understand whether EVO-induced cell death was attributed to ferroptosis, reactive oxygen species (ROS) production was detected by fluorescence microscopy with 2',7'-dichlorodihydrofluorescein diacetate (DCFDA) staining. TCCSUP cells were pretreated with the iron chelator deferoxamine (DFO) to maintain the homeostasis of intracellular iron ions. Figure 2A shows that ROS production was detected in the cells after exposure to EVO. However, the ROS level significantly decreased when cells were pretreated with DFO before EVO exposure. In contrast, cells treated with DFO alone had no impact on ROS production.“ We also revised the legend for Figure 2A as follows (lines 148-152): “(A) TCCSUP cells were treated with 100 µM of the iron chelator DFO or the vehicle (DMSO) for 1 h, followed by treatment with 15 µM EVO or the vehicle (DMSO) for 24 h. The cells were incubated with the ROS sensitive fluorometric probe DCFDA (10 μM) for 30 min in the dark, and the content of ROS was detected with fluorescence microscopy. Representative fluorescent microscopic images are shown (scale bar = 20 μM).” Accordingly, the text in the materials and methods section (4.4 Detection of cellular ROS production and lipid peroxidation) was also revised as follows (lines 372-378): “Cellular ROS generation was measured using the ROS sensitive fluorometric probe DCFDA (Invitrogen, Carlsbad, CA, USA). TCCSUP cells were seeded at 5 × 105 cells/dish in 10-cm dishes and cultured for 24 h. Subsequently, cells were pretreated with the iron chelator DFO (Cayman Chemical Company, Ann Arbor, MI, USA) or the vehicle DMSO for 1 h, followed by treatment with EVO or the vehicle DMSO for 24 h. The treated cells were stained with 10 μM DCFDA for 30 min at 37 °C in the dark. Finally, ROS-producing cells were detected under fluorescence microscopy (Olympus, Tokyo, Japan) for cells stained with green fluorescence.”

  1. Lines 117-119 – Define the use of C11-BODIPY as a lipid peroxidation stain to increase clarity

Response: Thank you for your valuable suggestion. We have revised the text in the results section as follows (lines 131-137): “As ferroptosis is driven by lethal accumulation of lipid peroxides in the cell membrane, determination of the lipid peroxide amount in cell membrane with the lipid peroxidation reporter probe C11-BODIPY and flow cytometry can be used to detect ferroptosis. Our results show that TCCSUP cells treated with EVO exhibited an increase in C11-BODIPY fluorescence, indicating that lipid ROS was generated, resulting in an increase in lipid peroxidation (Figure 2B-D). Furthermore, EVO-induced lipid peroxide accumulation could be alleviated by prior treatment with DFO (Figure 2B-D).” In addition, the text in the materials and methods section was also revised as follows (lines 379-382): “To detect membrane lipid peroxidation, TCCSUP cells that had been treated with DFO for 1 h and EVO for 24 h were incubated with 1 μM C11-BODIPY (Cayman Chemical) at 37 °C for 30 min in the dark. The amount of lipid peroxides was measured by flow cytometry (CytoFLEX, BD Biosciences, San Jose, USA).”

  1. Figure 2A – Define scale bar and increase size of it in the figure panel.

Response: Thank you for your kind reminder. We have added the scale bar in the image and defined the scale of the bar in the legend of Figure 2A (line 152). We also enlarged the size of Figure 2A to improve its clarity and visibility.

  1. Figure 2B and 2C – The authors need to confirm that there is no mistake in the selection of the data for these panels. They show close to the same results, it is possible the authors meant to show CTL and DFO alone in Figure 2B. In Panel C, the Y-axis title is visibly cropped.

Response: Thank you for pointing out our mistake. We have replaced the flow cytometry histogram shown in Figure 2B with the correct one. Levels of lipid peroxides are similar in DFO-treated and control (CTL) cells. We have also adjusted the image of Figure 2C to clearly show the Y-axis label.

  1. Line 127 – Authors should specify what are the indicated timepoints.

Response: Thank you for your helpful comment. We have revised the legend of Figure 2B-D as follows (lines 152-157): ”(B-D) TCCSUP cells were treated with DFO or DMSO for 1 h, followed by treatment of EVO or DMSO for 24 h as described in A. The cells were then stained with the lipid peroxidation reporter probe C11-BODIPY (1 μM) at 37°C for 30 min in the dark. The amount of lipid peroxides was measured by flow cytometry (n = 4 for each group). Data are mean ± SEM. CTL, control; ns, non-significant.”

  1. Figure 3 a. Define scale bar in panel A

Response: Thank you for your kind reminder. We have added the scale bar in the image and defined the scale of the bar in the legend of Figure 3A (line 193).

b. The authors should include viability data for DFO alone treatment.

Response: Thank you for your valuable suggestion. We have included the result of cell viability of DFO-treated TCCSUP cells in new Figure 3B. We also revised the text in the results section as follows (lines 169-171): “The cell viability did not differ between cells treated with DFO alone and untreated control cells, indicating that DFO is non-cytotoxic to TCCSUP cells (Figure 3B).”

  1. Figure 4  a. Panel A and B: The reduction in cell invasion may be confounded by cell death. The authors data in Figure 1B suggests a near 50% loss in viability of TCCSUP cells when treated with 30 µM EVO after 24 h, which is the same timepoint used for the migration assay. In addition, they only indicate a significant reduction in invasive capacity at 60 µM. My suggestion to the authors is to optimize the number of cells seeded for the transwell assay to allow for a longer period of measurement (48 and/or 72 h). This may allow for functional observations at lower EVO doses. However, viability assays need to be performed first in which, for example, 15 µM is shown to not significantly affect viability after 48 or 72 h.

Response: Thank you for highlighting this important question. To exclude the possibility that the effect of EVO on cell migration may be confounded by its cytotoxic effect, in particular, at high doses, we have modified the protocol of cell migration assay. We reduced the concentrations of EVO from 15 - 60 µM to 5 - 15 µM, as well as pretreated cells with EVO and washed out the drug before applying the treated cells to the transwell for assessing cell migration. We provided new data shown in Figure 4A and 4B. We revised the text in the results section (2.4. EVO inhibits cell migration and EMT in TCCSUP cells) as follows (lines 204-220): “In addition to ferroptosis, we also evaluated the effects of EVO on cell migration and EMT in TCCSUP cells. To exclude the possibility that the effect of EVO on cell migration may be confounded by its cytotoxic effect, in particular, at high doses, we used low concentrations of EVO (5 – 15 µM) to avoid evident cytotoxicity caused by EVO in the cell migration assay. We also pretreated cells with EVO and replenished with fresh medium before assessing its impact on cell migration. The capability of cell migration after exposure to EVO was assessed by the transwell migration assay. Figure 4A shows representative microscopic images of the migratory cells after pretreatment with various doses of EVO. Furthermore, quantitative analysis of migratory cells revealed that EVO inhibited cancer cell migration in a dose-dependent manner (Figure 4B). The cell viability of EVO-treated cells was concurrently assessed by the MTS assay. Figure S2A shows that EVO at concentrations of 10 and 15 µM only decreased cell viability by 10% and 15%, respectively, as compared to the control cells. Figure S2B shows the result of cell migration in EVO-treated TCCSUP cells after normalization of the migratory cell number with the percentage of cell viability. The degree of EVO-induced inhibition of cell migration remained similar after normalization of cell numbers (Figure S2A, B). There results clearly indicate that EVO inhibits cell migration independent of its cytotoxic effect.” In addition, we have revised the legend for Figure 4 as follows (lines 231-241): “(A, B) TCCSUP cells that had been treated with indicated concentrations of EVO for 24 h were seeded (1 × 104 cells) in the upper chamber of the transwell and cultured for 24 h. Cells that migrated through the membrane to the lower surface were stained with Giemsa and quantified. Representative microscopic images (magnification × 100; scale bar = 100 μM) (A) and quantitation (n = 8 for each group) (B) of the migratory cells are shown. (C-G) Detection (C) and quantitation of E-cadherin (D), N-cadherin (E), vimentin (F), and Snail (G) protein levels in TCCSUP cells after treatment with indicated concentrations of EVO for 24 h by western blot analysis (C) and densitometric analysis using the ImageJ soft-ware (n = 6 for each group) (D-G). Expression of β-actin served as the loading control. Data are mean ± SEM. The horizontal bars shown in D-G denote the mean value of each group.” Furthermore, we revised the text in the materials and methods section (4.3. Cell migration assay) as follows (lines 361-369): “Cell migration was measured using 24-well transwells with inserts containing 8-μm membrane filters (Merck Millipore, Burlington, MA, USA). TCCSUP cells were seeded at 5 × 105 cells/dish in 10-cm dishes and cultured for 24 h. Cells were then treated with EVO or the vehicle DMSO for an additional 24 h. The treated cells were then trypsinized and resuspended in the culture medium. Subsequently, the cell suspension (200 μl) containing 1 × 104 cells was added into the upper chamber of the transwells that were placed in 24-well plates to which the culture medium (600 μl) was added. After culture at 37 oC for 24 h, cells that migrated through the membrane to the lower surface were fixed, stained with Giemsa’s azur eosin methylene blue solution (Merck Millipore), and quantified.”

  1. Materials and Methods a. MTT Assay, lines 282 and 283. What do the authors mean by removing MTT reagent?
    1. The MTT assay is based on the production of insoluble formazan crystals by live cells, if these are disturbed or removed, it may confound the assay. Typically, a solvent is added directly to the wells after the cells are incubated in MTT reagent.

Response: We apologize for causing confusion about the MTT assay. We revised the text in the materials and methods section (4.2. Cell proliferation assay) as follows (lines 345-358): “Cell proliferation was determined to evaluate EVO-induced cytotoxic effect on TCCSUP cells using the MTT assay or MTS assay with the use of the CellTiter 96 AQueous One Solution cell proliferation assay kit (Promega, Madison, WI, USA) according to manufacturer’s instructions. TCCSUP cells were seeded at 5 × 104 cells/well in 96-well plates and cultured for 24 h. Cells were then treated with various concentrations of EVO (Sigma-Aldrich, St. Louis, MO, USA) for 24 h or with 30 μM EVO for different intervals of time. Subsequently, cells were replenished with fresh medium containing 1 mg/ml MTT (USB Corporation, Cleveland, OH, USA) or the combined MTS/PMS solution from the kit and then incubated at 37 °C for 2 h. For the MTT assay, culture medium with MTT was removed from the cells after a 2-h reaction. The formazan crystals left in the wells were subsequently dissolved in DMSO, and the plates were allowed to stand overnight at 37 oC. The optical absorbance at 570 nm (for MTT assay) or 490 nm (for MTS assay) was measured. The absorbance was then measured with a Multiskan GO microplate spectrophotometer (Thermo Scientific, Waltham, MA, USA).”

2. I recommend the authors to confirm their viability results with other viability assays to increase the robustness of their results.

Response: Thank you for your valuable recommendation. We also used the MTS assay, which is an aqueous soluble tetrazolium/formazan assay to measure cell viability after EVO treatment, as shown in Figure S1. We revised the text in the Results section (2.1. EVO induces cytotoxicity and promotes cell cycle arrest in TCCSUP cells) as follows (lines 95-102): “First, we evaluated the effects of EVO on the cell viability of human bladder cancer cells by the MTT assay. EVO induced cytotoxic effect in a dose- and time-dependent manner in TCCSUP cells (Figure 1A, B). We also used MTS, another tetrazolium compound, in the presence of PMS to assess EVO-induced cytotoxicity. An important advantage of MTS/PMS over MTT is the aqueous solubility of the reduced formazan product which eliminates the need for detergent solubilization or organic solvent extraction steps. Figure S1 shows EVO dose-dependently induced cytotoxic effect on TCCSUP cells with similar levels comparable to the result obtained from the MTT assay as shown in Figure 1A.”

Reviewer 2 Report

The methodology of experiments and explanation of results needs reworking for clarity and understandability. All figures need to be reworked, as they need confirmation of results, scales and more introduction. Materials and methods need to be expanded to better let the reader understand the decisions of why authors chose to do specific things, like removing MTT reagent, etc.

Author Response

The methodology of experiments and explanation of results needs reworking for clarity and understandability. All figures need to be reworked, as they need confirmation of results, scales and more introduction. Materials and methods need to be expanded to better let the reader understand the decisions of why authors chose to do specific things, like removing MTT reagent, etc.

Response: Thank you very much for your constructive comments. We have extensively revised our manuscript on the basis of your comments.

In Figures 1C, 2A, 3A, and 4A, we have added the scale bars in the images and defined the scale of the bar in their figure legends.

In Figure 2, we have provided correct new flow cytometry histograms for Figure 2B and improved the image quality of Figure 2C. Based on the two figures provided, we conclude that treatment with DFO alone does not have a significant effect on the content of lipid peroxides compared to the control group. However, treatment with EVO increases the level of lipid peroxides, and such effect can be reversed if cells are pretreated with DFO.

In Figure 3B, we have added cell viability data for cells treated with DFO alone. The result reveals that DFO treatment has no impact on cell viability.

For cell migration assay, we have provided new data for Figure 4A, 4B, S2A, and S2B to exclude the possibility that the effect of EVO on cell migration may be confounded by its cytotoxic effect, in particular, at high doses. Please refer to our response to Reviewer 1 (Question 8).

In Figure 5H, we present a schematic illustration of how EVO induces ferroptosis in human bladder cancer cells through suppression of GPX4.

In the results section, we have clarified that we used DCFDA staining to determine the reactive oxygen species (ROS) production. Besides, we used the lipid peroxidation reporter probe C11-BODIPY staining and flow cytometry to assess lipid peroxide content (lines 124-137). We have provided detailed information about the timepoints of each treatment, as well as relevant staining methods and duration for each experiment in figure legends (lines 148-157). We also revised the text in the materials and methods section accordingly (lines 372-382).

We have also extensively revised the text in materials and methods section according to the comments raised by the reviewers (lines 344-389). Please also refer to our responses to Reviewer 1 (Questions 2, 3, 8, and 9).

Regarding the English language and style of our manuscript, we engaged the editing services provided by the journal to ensure thoroughness in identifying and correcting any errors. In addition, the manuscript was also extensively edited by a senior colleague.

Reviewer 3 Report

In the present manuscript the authors have shown anti cancer activity of Evodiamine in a bladder cancer model. There is lot of work done on this molecules and considering the previous work the study lacks novelty and the rationality of each experiment is not properly understood from the manuscript.

Author Response

In the present manuscript the authors have shown anti-cancer activity of Evodiamine in a bladder cancer model. There is lot of work done on this molecule and considering the previous work the study lacks novelty and the rationality of each experiment is not properly understood from the manuscript.

Response: Thank you very much for agreeing with us to the intention of this manuscript. We have carefully revised our manuscript to improve the clarity and  understandability and to emphasize the importance of our findings. This is the first study to demonstrate that evodiamine induces cell death through the process of ferroptosis in bladder cancer cells. We also proposed a mechanism of action that evodiamine induces ferroptosis via suppression of the anti-oxidant enzyme GPX4, and that the increased lipid peroxidation following evodiamine treatment can be reversed by the iron chelator DFO.

In Figure 1, we confirmed that evodiamine induces cell death of bladder urothelial carcinoma. To the best of our knowledge so far, only two studies have been reported to investigate the effects of evodiamine on urothelial carcinoma cells.  
    Reference 15: Zhang, T. et al; Evodiamine induces apoptosis and enhances TRAIL-induced apoptosis in human bladder cancer cells through mTOR/S6K1-mediated downregulation of Mcl-1; Int J Mol Sci. 2014 Feb 21;15(2):3154-71.
    Reference 16: Shi, C.S. et al; Evodiamine induces cell growth arrest, apoptosis and suppresses tumorigenesis in human urothelial cell carcinoma cells; Anticancer Res. 2017 Mar;37(3):1149-1159.

We have revised the introduction section of our manuscript to state that there is a need for further investigation into the anti-cancer effect and the mechanism of evodiamine in bladder urothelial carcinoma (lines 65-66): “However, there is currently limited literature discussing the anti-cancer effects of evodiamine in bladder UCs [15, 16] .“

In addition, in response to the comment of reviewer 1 (Question 9a), we have conducted the MTS assay for assessing EVO-induced cell death, as shown in Figure S1, to support the observation of the dose-dependent decrease in cell viability following evodiamine treatment. Please refer to our response to reviewer 1 (Question 9a).

In Figure 2, our objective was to demonstrate EVO-induced cytotoxicity through ferroptosis, which was associated with elevated levels of cellular ROS and lipid peroxides. Please refer to our response to reviewer 1 (Questions 2 and 3). We hope that our revision will help to clarify the rationale of our experiments and make our manuscript more easily understood.

In Figure 3, we aimed to confirm that the observed cell death induced by EVO was indeed attributed to ferroptosis. We confirmed that the iron chelator DFO effectively reverses EVO-induced cell death, whereas apoptosis or necroptosis inhibitors have no effects. In Figure 3B, we have added the cell viability data from cells treated with DFO alone. The cell death caused by DFO treatment is similar to that of the control group. We have added related description in the results section as follows (lines 169-171): “The cell viability did not differ between cells treated with DFO alone and untreated control cells, indicating that DFO is non-cytotoxic to TCCSUP cells (Figure 3B).”

In Figure 4, we illustrate that EVO can inhibit cancer cell migration. Please refer to our response to reviewer 1 (Question 8).

In Figure 5, we used human TCCSUP bladder tumor xenograft model to verify the effect of EVO on inhibition of tumor growth and EMT. In addition, we present a schematic illustration of how EVO induces ferroptosis in bladder cancer cells by suppression of GPX4, and how the iron chelator DFO reverses such effect by chelating iron ions (Figure 5H). We have also added the conclusion of our study in the abstract section to emphasize our novelty: “In conclusion, EVO is a novel inducer to activate ferroptosis of bladder cancer cells and may be a potential therapeutic agent for bladder cancer.(lines 45-46)

Overall, on the basis of our responses to the reviewers, we believe that our manuscript has been greatly improved towards IJMS standards after revision.

Round 2

Reviewer 1 Report

1. Authors have a minor mistake when defining the scale bars. The definition should be in µm and not µM. 

2. Authors claim that the MTT and MTS cell viability assays report similar results. However, when comparing Figure S1 to Figure 1A, the dose-response curve on Figure S1 is steeper, indicating a higher degree of cytotoxicity. Unless the authors have any explanation to this disparity, I am inclined to believe that either the MTT or MTS assays were not conducted properly. Perhaps, the MTT assay, due to the extraction of the MTT reagent out of the wells as opposed to solubilizing the formazan pellets within the wells in the initial experiments, which was highlighted in the first revision. The authors need to consider that the MTT viability data is not correct and repeat any needed experiments using the MTS assay. 

3. Authors claim that there is only a small loss to cell viability when TCCSUP cells are treated with 5-10 µM of EVO. However, Figure S1 indicates a viability of ~30% at 10 µM after 24 h treatment with EVO. Based on the results of Figure S1, the results in Figure S2 panel A do not match up. The authors need to clarify if the conditions of Figure S1 and Figure S2 panel A are different in any aspect. 

Author Response

  1. Authors have a minor mistake when defining the scale bars. The definition should be in µm and not µM.

Response:

Thank you for pointing out our mistake. These mistakes have been corrected in the legends for Figure 1 (page 4, line 120), Figure 2 (page 5, line 162), Figure 3 (page 7, line 205), and Figure 4 (page 9, line 250).

  1. Authors claim that the MTT and MTS cell viability assays report similar results. However, when comparing Figure S1 to Figure 1A, the dose-response curve on Figure S1 is steeper, indicating a higher degree of cytotoxicity. Unless the authors have any explanation to this disparity, I am inclined to believe that either the MTT or MTS assays were not conducted properly. Perhaps, the MTT assay, due to the extraction of the MTT reagent out of the wells as opposed to solubilizing the formazan pellets within the wells in the initial experiments, which was highlighted in the first revision. The authors need to consider that the MTT viability data is not correct and repeat any needed experiments using the MTS assay.

Response:

Thank you for your helpful suggestion. Indeed, the MTS assay was shown to be more accurate and simpler than the MTT assay in terms of the correlation with cell density, viability, and proliferation of hepatocytes [1]. MTS showed better correlation coefficients with tritiated thymidine incorporation than MTT for assessing cell viability and proliferation. In evaluating anti-parasite chemotherapy, the MTS method was more reproducible and displayed higher sensitivity compared with the MTT method [2]. Another advantage of MTS over MTT is that the MTS assay requires no washing or cell harvesting, thus eliminating solubilization steps to dissolve formazan crystals normally required for the MTT assay [3]. Therefore, we have repeated the cell viability assays using the MTS method and provided new data in Figures 1A, 1B, and 3B to improve data quality. Furthermore, we included the following in the results section (page 2, lines 96-101):

“ First, we evaluated the effects of EVO on the cell viability of human bladder cancer cells with the tetrazolium salt MTS and the electron coupling reagent phenazine ethosulfate (PES). The MTS formazan product is soluble in tissue culture medium, which eliminates solubilization steps normally required for the MTT assay in which insoluble formazan crystals are formed [33]. The colorimetric MTS assay shows that EVO induced cytotoxic effect in a dose- and time-dependent manner in TCCSUP cells (Figure 1A, B).” 

References:

  1. Wang L, Sun J, Horvat M, Koutalistras N, Johnston B, Sheil A.G.R. Evaluation of MTT, XTT, MTT and 3HTdR incorporation for assessing hepatocyte density, viability and proliferation. Methods Cell Sci 1996, 18: 249-255.
  2. Henriques C, Moreira TLB, Maia-Brigagao C, Henriques-Pons A, Carvalho TMU, de Souza W. Tetrazolium salt based methods for high-throughput evaluation of anti-parasite chemotherapy. Methods 2011, 3:2148-2155.
  3. Cory AH, Owen TC, Barltrop JA, Cory JG. Use of an aqueous soluble tetrazolium/formazan assay for cell growth assays in culture. Cancer Commun. 1991, 3: 207-212.

  1. Authors claim that there is only a small loss to cell viability when TCCSUP cells are treated with 5-10 µM of EVO. However, Figure S1 indicates a viability of ~30% at 10 µM after 24 h treatment with EVO. Based on the results of Figure S1, the results in Figure S2 panel A do not match up. The authors need to clarify if the conditions of Figure S1 and Figure S2 panel A are different in any aspect.

Response:

We apologize for the lack of clarity. Since we replaced old cell viability data assessed by the MTT assay with new data determined by the MTS assay (Figure 1A, 1B, and 3B), we deleted old Figure S1 that is similar to new Figure 1A. In fact, in both new Figure 1A and old Figure S1, cell viability was assessed immediately after EVO treatment for 24 h. However, in new Figure S1A (i.e. old Figure S2A), in order to detect the viability status of cells during migration, we used the same cell samples that had been treated with EVO for 24 h and replenished with fresh culture medium in the absence of EVO for detecting the viability and migration of cells after being cultured for an additional 24 h. Since treatment conditions, including the duration for EVO treatment and the assay time point, were different between the EVO dose-dependent experiment (new Figure 1A and old Figure S1) and the viability status experiment for cells undergoing migration assay shown in Figure S1 (i.e. old Figure S2), it would be inappropriate to compare the cell viability shown in Figure 1A and Figure S1A. We have also amended the sentences to describe the protocols more clearly for concurrent detection of cell migration and cell viability after EVO treatment (page 8, lines 220-226):

“TCCSUP cells cultured in 10-cm culture dishes were treated with or without EVO for 24 h. The treated cells were then trypsinized, resuspended in the culture medium in the absence of EVO, reseeded at 1 × 104/well in the upper chamber of the transwell in 24-well plates and in 96-well plates, and cultured for an additional 24 h for assessing cell migration and cell viability, respectively. Figure 4A shows representative microscopic images of the migratory cells with or without EVO pretreatment. The number of migratory cells decreased in EVO-treated cells compared with that in the control cells.”   

We have also revised the legend for Figure S1 as follows:

“(A, B) TCCSUP cells cultured in 10-cm culture dishes were treated with or without EVO for 24 h. The treated cells were then trypsinized, resuspended in the culture medium in the absence of EVO, reseeded at 1 × 104/well in 200 µl culture medium in 96-well plates (A) and in the upper chamber of the transwell (B), and cultured for an additional 24 h for assessing cell viability with the MTS assay (n = 8) (A) and cell migration with the transwell migration assay (B). Cells that migrated through the membrane to the lower surface were stained with Giemsa and quantified (n = 8) (B).”

Reviewer 2 Report

The article is now significantly improved. However, in the introduction or discussion, I would include a small section about bladder cancer and the effects of non-coding RNA, regulation of ferroptosis and possible effects on Evodiamine use/effectivity. As ncRNAs are now studied in association to treatment, diagnosis and more aspects of diseases, the article could benefit from this insight.

Suggested articles on ferroptosis and noncoding RNAs:

https://doi.org/10.1038/s41418-022-00998-x

https://doi.org/10.3389/fmolb.2022.1003045

DOI: 10.3390/ijms232113206

Author Response

The article is now significantly improved. However, in the introduction or discussion, I would include a small section about bladder cancer and the effects of non-coding RNA, regulation of ferroptosis and possible effects on Evodiamine use/effectivity. As ncRNAs are now studied in association to treatment, diagnosis and more aspects of diseases, the article could benefit from this insight.

Suggested articles on ferroptosis and noncoding RNAs:

https://doi.org/10.1038/s41418-022-00998-x

https://doi.org/10.3389/fmolb.2022.1003045

DOI: 10.3390/ijms232113206

Response:

We are grateful for your constructive comment. We have included the following as the fourth paragraph of the discussion section to address this issue (pages 12-13, lines 351-384):

“Non-coding RNAs, such as microRNA (miRNA) and long non-coding RNA (lncRNA), play pivotal roles in gene regulation and biological processes in both health and disease, including cancer initiation and progression. Therefore, they are recognized as potential diagnostic and prognostic markers as well as therapeutic targets for cancers, including bladder UC [44]. Increasing evidence has indicated that ferroptosis is associated with cancer initiation, progression, and suppression [45]. Furthermore, miRNAs can either promote or suppress tumors through regulating ferroptosis by interfering with iron metabolism, lipid peroxidation, and the system Xc--GPX4 axis [46, 47]. EVO can affect the expression of non-coding RNAs and their target genes in cancer cells, suggesting that EVO can mediate gene expression at the post-transcriptional level. Accumulating evidence has revealed that miRNAs are involved in the control of DNA methylation machinery [48-50]. Huang et al. reported that EVO increases the expression of miR-152, miR-429, and miR-29a, which in turn downregulates DNA methyltransferase 1 (DNMT1), DNMT3A, and DNMT3B in colorectal cancer cells, resulting in reversing epigenetic silencing of tumor suppressor genes and inhibiting cancer cell growth [51]. Apart from DNA methylation, the three miRNAs possess functions as tumor suppressors. In human hepatocellular carcinoma tissues, miRNA-152 levels are reduced and transferrin receptor 1 (TFR1) is over-expressed. Furthermore, miR-152 regulates iron homeostasis by downregulating TFR1, thereby participating in ferroptosis in liver cancer cells [52]. The miR-200 family is highly expressed in epithelial cells and plays a crucial role in maintaining the epithelial phenotype, which is achieved by inhibiting the expression of factors that promote EMT [53]. MiR-429 belongs to the miR-200 family, suggesting that it may have roles in EMT sup-pression [53, 54]. Moreover, miR-29a inhibits EMT in lacrimal gland adenoid cystic carcinoma and endometrial cancer [55, 56]. LncRNAs have been regarded as one of the key regulators in cancer progression and drug resistance by regulating the expression of downstream genes and various biological processes. Silencing of lncRNA NEAT1 can enhance erastin-mediated ferroptosis in non-small-cell lung cancer cells by increasing intracellular lipid peroxidation, indicating the involvement of NEAT1 in regulation of ferroptosis sensitivity [57]. NEAT1 plays a tumor-promoting role in ovarian cancer. It was shown recently that EVO decreases the expression of NEAT1 and CDK19, but increases miR-152 expression in ovarian cancer cells [58]. Furthermore, EVO attenuates ovarian cancer cell progression via the NAET1-miR-152-3p-CDK19 axis [58]. In conclusion, EVO can upregulate tumor suppressor miRNAs and down-regulate oncogenic lncRNA NEAT1, providing molecular mechanisms linking EVO to non-coding RNAs that can contribute, in part, to its anti-cancer activities.”

Reviewer 3 Report

The authors have made significant improvements of the article.

The authors should improve figure 2A. The figure's quality is not up to the standard due to high background fluorescence. The authors should also use a counterstain like DAPI.

Author Response

The authors have made significant improvements of the article.

Response:

We thank the reviewer’s positive comment.

The authors should improve figure 2A. The figure's quality is not up to the standard due to high background fluorescence. The authors should also use a counterstain like DAPI.

Response:

Thank you for your valuable suggestion. To improve data quality of Figure 2A, we have repeated the experiments and presented new data with fluorescent (green staining for ROS and blue staining for the nucleus) and bright-field microscopic images. The cells were also counterstained with DAPI as suggested. 

Round 3

Reviewer 1 Report

The authors have greatly improved the quality of the article. The authors addressed all my concerns with the cellular cytotoxicity experiments and now their results support their conclusions more strongly.